# Novel multicellular prokaryote discovered next to an underground stream

Kouhei Mizuno[1,2]*, Mais Maree[3], Toshihiko Nagamura[2], Akihiro Koga[2], Satoru Hirayama[4,5], Soichi Furukawa[4†], Kenji Tanaka[6], Kazuya Morikawa[7]*

[1]Division of International Affairs, Headquaters, National Institute of Technology, Tokyo, Japan; [2]Department of Creative Engineering, National Institute of Technology, Kitakyushu, Japan; [3]Doctoral Program in Biomedical Sciences, Graduate School of Comprehensive Human Sciences, University of Tsukuba, Tsukuba, Japan; [4]Department of Food Bioscience and Biotechnology, College of Bioresource Sciences, Nihon University, Fujisawa, Japan; [5]Division of Microbiology and Infectious Diseases, Graduate School of Medical and Dental Sciences, Niigata University, Niigata, Japan; [6]Department of Biological and Environmental Chemistry, School of Humanity-Oriented Science and Engineering, Kindai University, Iizuka, Japan; [7]Division of Biomedical Science, Faculty of Medicine, University of Tsukuba, Tsukuba, Japan

**Abstract** A diversity of prokaryotes currently exhibit multicellularity with different generation mechanisms in a variety of contexts of ecology on Earth. In the present study, we report a new type of multicellular bacterium, HS-3, isolated from an underground stream. HS-3 self-organizes its filamentous cells into a layer-structured colony with the properties of a nematic liquid crystal. After maturation, the colony starts to form a semi-closed sphere accommodating clusters of coccobacillus daughter cells and selectively releases them upon contact with water. This is the first report that shows that a liquid-crystal status of cells can support the prokaryotic multicellular behavior. Importantly, the observed behavior of HS-3 suggests that the recurrent intermittent exposure of colonies to water flow in the cave might have been the ecological context that cultivated the evolutionary transition from unicellular to multicellular life. This is the new extant model that underpins theories regarding a role of ecological context in the emergence of multicellularity.

*For correspondence:
k_mizuno@kosen-k.go.jp (KM);
morikawa.kazuya.ga@u.tsukuba.ac.jp (KM)

†Deceased

**Competing interest:** The authors declare that no competing interests exist.

## Editor's evaluation

This article reports the discovery of an unusual form of bacterial multicellularity – an organism that can exist in dense, filamentous multicellular structures and clusters of coccobacillus daughter cells. Experiments that mimic the periodic immersion that the bacteria experience in their natural cave environment suggest that water immersion plays a role in these life-cycle dynamics. This work, while rather qualitative, will likely attract great interest from a diverse range of scientists working on multicellularity, the biophysics of cell packing, and geobiological problems.

## Introduction

The emergence of multicellularity is one of the mysteries of life on Earth, and has been attracting scientists in a broad range of fields including bioengineering on artificial construction of cellular behaviors (*Basu et al., 2005*) and medical research on cancer progression (*Trigos et al., 2017*). The oldest reliable fossil record of multicellular life is an ~3.42-billion-year-old putative filamentous microfossil that inhabited a paleo-subseafloor hydrothermal vein system (*Cavalazzi et al., 2021*). Since then, the fossil records show the emergence of dominating multicellular cyanobacteria on this planet

## Aggregative Multicellularity
### Social aggregation

### Aggregation

Some bacteria combine motility with intercellular communication, resulting in social behavior or fruit-body formation,   e.g., *Myxococcus*.

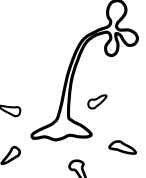

## Multicellular-like behavior
### Differentiation & Subpoplulation

Bacteria can differentiate. e.g., *Bacillus* sporulation, *Caulobacter* stalk formation. Furthermore, bacteria can generate subpopulations. e.g., phase variation, bistability (Casadesús and Low, 2013; Dubnau and Losick, 2006). These are also examples of multicellularity that benefit the population as a whole (Malik and Rosenberg, 2009).

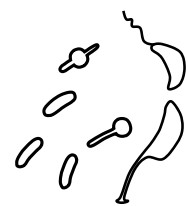

## Clonal Multicellularity   Serial Cell Division

### Filamentation

Filamentation is perhaps the oldest form of multicellularity on Earth. Fossil records show that photosynthetic cyanobacteria existed more than 2 billion years ago (Schirrmeister et al., 2011), e.g., *Anabaena*.

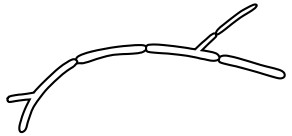

### Biofilm formation

Biofilms can exhibit functional coordination like multicellular organisms (Nadell and Bassler, 2011). e.g., *Pseudomonas*, *Vibrio cholerae*.

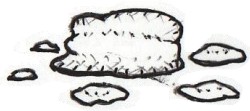

### Patterned multicellularity

Some colonies self-organize into a structural patterned colony (Shapiro, 1998); this is termed 'Patterned multicellularity' (Claessen et al., 2014).
e.g., *Streptomyces*, *Bacillus*, *Pseudomonas*, *Proteus mirabilis*, etc.

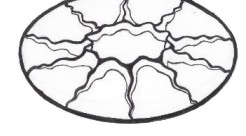

### Multicellular magnetotactic prokaryotes

Found in intertidal zones, like lagoons. These organisms are typically composed of 10–100 cells and are known to accumulate iron within cells (Zhou et al., 2013).

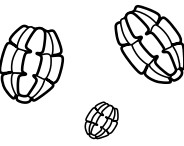

**Figure 1.** Prokaryotic multicellularity. 'Clonal multicellularity', 'aggregative multicellularity' (*Brunet and King, 2017*), and 'multicellular-like behavior' (*Malik and Rosenberg, 2009*) describe modes of multicellularity that arose through distinguished hypothetical conditions.

(*Amard and Bertrand-Sarfati, 1997*; *Golubic and Seong-Joo, 2004*), which were identified as early as 2.32 billion years ago and are assumed to be responsible for the rapid accumulation of oxygen levels, known as the 'Great Oxygenation Event' (*Schirrmeister et al., 2011*). It seems that multicellularity emerged several times independently at different species lineages irrespective of a eukaryotic or prokaryotic phenotype (*Brunet and King, 2017*). Multicellularity in prokaryotes would not directly link to the origin of animals, but investigating it gives us fundamental models to consider the mechanistic or evolutionary steps behind the leap to multicellularity. Currently, we see a diversity of prokaryotes that exhibit multicellularity with distinct formation mechanisms in different contexts of ecology on Earth (*Claessen et al., 2014*; *Lyons and Kolter, 2015*; *Nadell and Bassler, 2011*; *Vlamakis et al., 2013*; *Zhou et al., 2013*; *Figure 1*).

Actinomyces and cyanobacteria are well-studied prokaryotes that exhibit multicellular forms (*Claessen et al., 2014*). Their reproducible multicellular structures achieve a high fitness for the whole

population even though they are not beneficial for partial individuals. Further elaborate organization can be seen in large-sized genome bacteria, such as *Myxobacteria* (9–16 Mbp genome) and *Streptomyces* (7–12 Mbp) (*Claessen et al., 2014*). Some fitness advantages acquired by multicellularity include avoiding predation (*Bernardes et al., 2021*; *Boraas et al., 1998*; *Herron et al., 2019*), cooperative defense (*Shapiro, 1998*), dispersal (*Smith et al., 2014*), labor division (*Brunet and King, 2017*; *Shapiro, 1998*; *Koschwanez et al., 2011*), resource competition (*Herron et al., 2019*), and creation of a stable internal environment (*Brunet and King, 2017*).

The terms 'aggregative multicellularity' or 'clonal multicellularity' are used to describe the different bottom-up processes resulting in multicellular structures (*Figure 1*). An example of aggregative multicellularity can be seen in myxobacteria, where solitary cells come together upon stress conditions to organize a fruiting body (*Bonner, 1998*). On the other hand, clonal multicellularity occurs through incomplete cell division, such as in actinomyces and cyanobacteria, and may be the process by which the first eukaryotic ancestor of all animals evolved (*Brunet and King, 2017*). The co-option of preexisting cellular traits is an important concept in these bottom-up processes; for example, cell aggregation in response to stress, or a sticky cellular trait might have been used to architect a multicellular status (*Brunet and King, 2017*; *Malik and Rosenberg, 2009*).

Once a multicellular organism has emerged, the multicellular trait needs to be internalized as genetic information and fixed in the population for which Darwinian natural selection, which occurs over a longer timescale encompassing multiple cell lifetimes, must take place (*Black et al., 2020*). This is a key problem of the bottom-up perspective of multicellularity (*Libby and Ratcliff, 2014*). However, the constraints that allow for the selection of a multicellular status are as yet unknown. One suggested explanation of the fixation of multicellularity is the involvement of certain environmental factors that can give an extrinsic selective pressure to the evolving population. This concept is termed 'Ecological Scaffolding' (*Black et al., 2020*) and has been experimentally substantiated in a model experiment using *Pseudomonas* (*Hammerschmidt et al., 2014*). The study regarded a biofilm formed at the air–liquid interface of a glass bottle as a population at a certain generation, and counted repeated dynamism of synthesis and collapse of the biofilm as generations of the population. This concept suggests that Darwinian natural selection is possible even for unicellular organisms through support by external environmental constraints before multicellular traits are fixed in a population.

The cave microbiome was previously considered to simply be a subset of the surface microbes (*Laiz et al., 1999*), but it has recently been shown that the overlap of operational taxonomic units with microbes from the surface is only 11–16% (*Lavoie et al., 2017*; *Ortiz et al., 2013*). Furthermore, the microbiomes of different cave environments differ, as has been revealed in caves composed of limestone (*Barton and Northup, 2007*; *Ortiz et al., 2013*, *Rangseekaew and Pathom-Aree, 2019*), lava (*Lavoie et al., 2017*), ice (*Itcus et al., 2018*), quartz (*Sauro et al., 2018*), or sulfur (*Macalady et al., 2008*). Namely, each cave has a distinctive microbiome that has adapted to that environment.

Here, we report a new cave bacterium that can develop a multicellular-like architecture, possibly through the influence of an external environmental constraint, from a simple, but ordered, cell cluster. The bacterium was isolated from the surface of a cave wall above an underground river in a limestone cave system that is intermittently submerged. The most remarkable aspect of this bacterium's lifecycle is the existence of well-regulated development (dimorphism) between the first growth stage (filamentous cells that show liquid-crystal-like self-organization on a solid surface) and the second stage (coccobacillus cells that can disperse in flowing water). There is no report that the properties of a nematic liquid crystal were co-opted toward prokaryotic multicellularity. The second stage suggests the involvement of recurrent water flows in the establishment of the multicellularity in this species. We discuss what the discovery of HS-3 implicates in the context of our present understanding and hypothesis about the emergence of multicellularity.

## Results

### *Jeongeupia sacculi* sp. nov. HS-3, a new bacterial species isolated from a karst cave

We previously collected oligotrophic bacteria from a variety of environments and studied their physiology (*Ilham et al., 2014*; *Mizuno et al., 2017*; *Mizuno et al., 2010*). In 2008, strain HS-3 was isolated from water dripping on a limestone cave wall in the Hirao karst plateau (*Fukuyama et al.,*

*2004*) in northern Kyushu Island, Japan (*Figure 2—figure supplement 1*). The sampling site, which was located slightly above the water surface of an underground river, was intermittently submerged due to an increase in runoff after rainfall (*Figure 2A*, *Figure 2—figure supplement 1*). After plating on agar, colonies appeared transparent and iridescent (*Figure 2B*). HS-3 is a Gram-negative obligate aerobe with a single polar flagellum (*Figure 2F*, inset). Genetically, it belongs to a group in the family *Neisseriaceae* that is mainly comprised of environmental bacteria (*Figure 2—figure supplement 2*). It has a 3.4 Mbp genome and a plasmid (2.0 kbp). Based on phenotypic comparisons with other closely related species (*Supplementary file 1*), HS-3 is considered to be a novel species, and we name this species as *Jeongeupia sacculi* sp. nov. HS-3. Its optimal growth temperature (24°C) and pH (8.0–9.0) are consistent with conditions in the cave. Another species in the same group that is adapted to life in oligotrophic environments is *Deefgea rivuli*, which has been isolated from tufa, a form of limestone (*Stackebrandt et al., 2007*; *Figure 2—figure supplement 2A*). However, no reports of distinctive colony morphology or cell differentiation have been reported in this group to date.

## HS-3 forms colonies with a liquid-crystal texture

Bacterial cells capable of growth on agar medium typically proliferate in a disordered state, yielding colonies with an opaque texture. HS-3 formed transparent colonies with an iridescent hue (*Figure 2B*). Specifically, the colonies assumed an anisotropic optical structure (*Figure 3E*), suggesting that the cells were orientated in a manner similar to liquid-crystal molecules. As shown in *Figure 2C–E*, the cells in the colonies formed vortex-shaped structures composed of closely arrayed filamentous cells with bulges on the surface. Transforming these cells with the green fluorescent protein (GFP) gene revealed that these structures were composed of cells rather than some extracellular matrix or artifacts derived from electron microscopy (*Figure 2G*). On a solid medium, the length of the cells was correlated with the colony density, and lower densities were associated with a wider range of cell length (*Figure 2F*). These findings suggest that cell elongation in this strain is a regulated process that occurs in response to environmental cues. Cell elongation was not observed in liquid cultures.

A time course analysis of the growing colonies showed that the cells proliferate as coccobacilli for the first 10 hr (data not shown). Then, as cell elongation was initiated from the edge of the colony, the cells gradually assumed topological characteristics similar to the two-dimensional nematic pattern described in liquid-crystal theory (*Doostmohammadi et al., 2018*; *Duclos et al., 2017*), with comet-like (+1/2) and trefoil-like (−1/2) topological defects being visible (*Figure 3—figure supplement 1*). Generally, the microcolonies initially spread as a single layer. As the colony expands, bulges are produced especially, but not specifically, around the rapidly growing colony edge (*Figure 2E*). We consider that the morphological change occurs to sustain the ordered state by relieving internal physical pressure. The disturbed single layer at the center then serves as the focal point for the expansion of additional layers (*Figure 3A*, differential interference contrast). The bulges are also generated around the colony center (data not shown). As the multilayered colony continues to grow, the expanding edge swallows the adjacent region and the internal filamentous cells buckle, generating domains with a vortex structure (*Figure 3F*, *Video 1*). Importantly, the filamentous cells appear to sustain the well-aligned structure, suggesting that there is tight adhesion among the filamentous cells (*Figure 2E*). An anisotropic pattern that emerged on the bottom layer of the colony was sustained throughout the period of colony growth (*Figure 3C, D*). Furthermore, the optical characteristics of the multilayer colonies indicate that the ordered structures are sustained (*Figure 3E*).

## Internal proliferation to germ-like phase

Colonies stopped growing at 2 days after inoculation and no further changes were observed for the following 2–3 days. However, internal proliferation then started and the colony began to swell three dimensionally (*Video 2*). Internally proliferating cells formed optically distinct regions within a transparent colony, suggesting the loss of the ordered-layer structure. Nile red, a lipid-staining fluorescent dye, was used to selectively stain the membranes of the elongating cells in a colony (*Strahl et al., 2014*), and the internal coccobacillus cells were recognized as nonstained areas in three-dimensional cross-sectional images (*Figure 4A*), as well as in micrographs obtained by conventional fluorescence microscopy (*Figure 4B*). At the fifth day after inoculation, the internal cells were crowded out of the colony in our experimental setting (*Figure 4B*, right panels). The occurrence of this event in a single

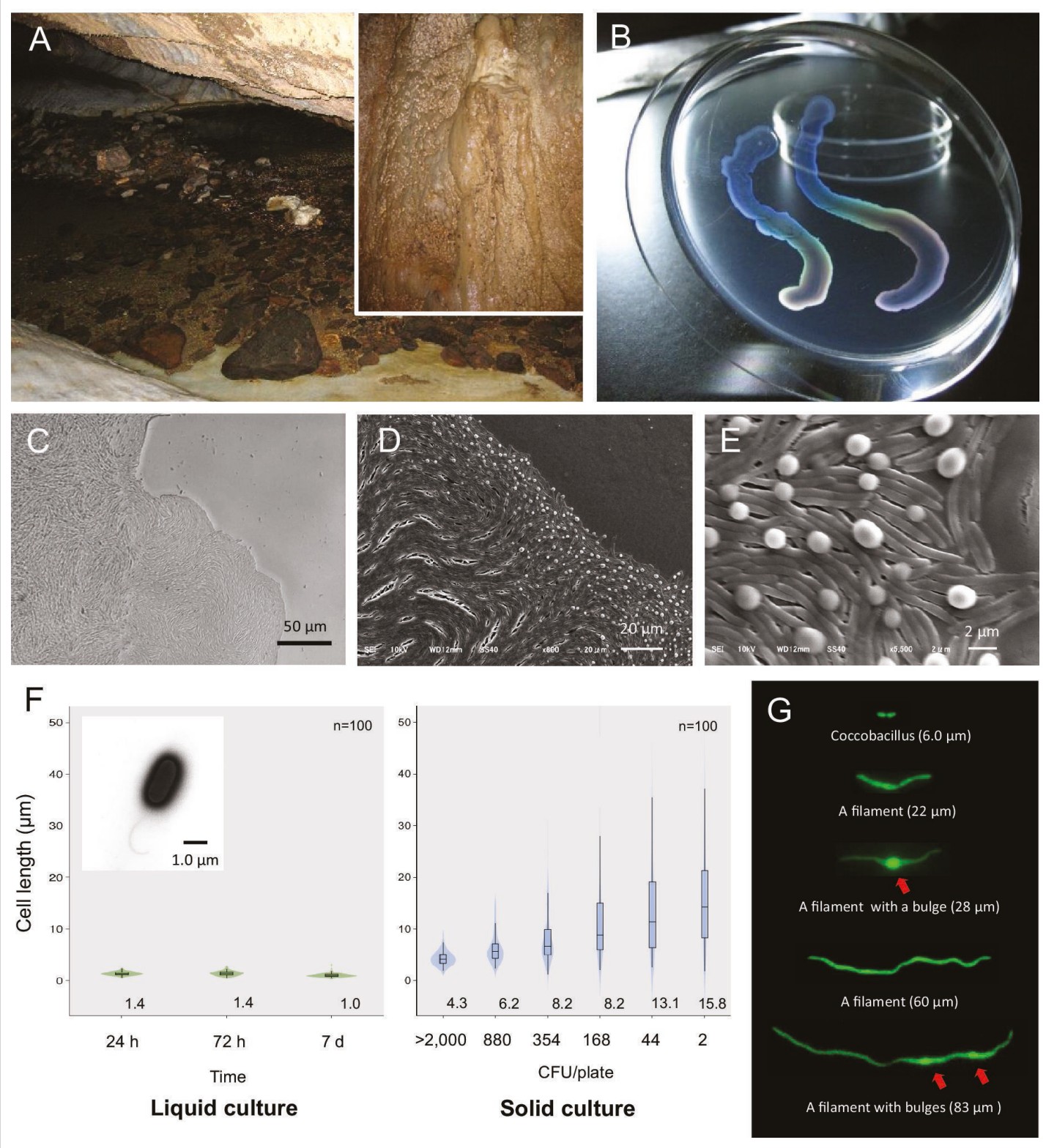

**Figure 2.** Liquid-crystal appearance and microstructure of a colony of *Jeongeupia sacculi* sp.nov. HS-3, discovered in a limestone cave. (**A**) River in a cave showing the sampling site on the cave wall (inset). (**B**) Transparent and iridescent appearance of colonies. The edge of a colony growing on agar. Differential interference contrast (DIC) (**C**) and scanning electron microscopic (SEM) (**D, E**) images. Bulges of cells on the colony surface (**E**). (**F**) Cell length in liquid and solid cultures. The liquid cultures were monitored at different times. The solid cultures were inoculated at different colony densities (CFU/plate) and cultured for 2 days. The value given for each sample is the mean obtained from 100 cells. Inset: transmission electron microscopic

*Figure 2 continued on next page*

*Figure 2 continued*

(TEM) image of a liquid-cultured cell. (**G**) Typical cell morphology from a colony cultured on agar and observed by green fluorescent protein (GFP) fluorescence. Arrows: bulges of cells.

The online version of this article includes the following figure supplement(s) for figure 2:

**Figure supplement 1.** The isolation site of strain HS-3.

**Figure supplement 2.** Phylogenetic analysis and genome information of HS-3.

colony triggered a chain reaction of this phenomenon in adjacent colonies (*Figure 4C*), indicating some form of control.

The sampling site of HS-3 was a moist cave wall above an underground river (*Figure 2A*). Since this site is also intermittently submerged by water due to increased runoff caused by rainfall, we examined the response of colonies on an agar plate to being submerged by water. Remarkably, the coccobacilli were released into the water column, leaving the architecture produced by the filamentous cells behind (*Figure 4D*, *Video 3*). Even after the complete release of the coccobacillus from the center of the colony, the bottom cell layer of the aligned filamentous cells adhered tightly to each other (*Figure 4D*, right side). These observations suggest that the filamentous cells act to support the embedded coccobacilli clusters. We also examined whether the released 'daughter' cells were able to reproduce the original colony morphology. The results showed that the spawned coccobacillus cells could indeed reproduce the original filamentous colony morphology (*Figure 4—figure supplement 1A*). In addition, when filamentous cells obtained from a mature colony were inoculated on agar (*Figure 4—figure supplement 1B*) the majority of cells failed to regrow, potentially due to cell damage during recovery. However, a fraction elongated and then divided to generate short rods that each formed a microcolony (*Figure 4—figure supplement 1B*). Thus, the filamentous cells were not irreversibly destined to be the equivalent of 'somatic' cells. These results suggest that the different morphological changes observed in the life-cycle of HS-3 are reversible (*Figure 4E*).

## Discussion

There has been a wide variety of work regarding the emergence of multicellularity on extant multicellular organisms such as volvocine green algae and holozoans (e.g., choanoflagellates and ichthyosporeans) (*Bonner, 1998*; *Brunet and King, 2017*; *King et al., 2008*; *Matt and Umen, 2016*; *Sebé-Pedrós et al., 2016*). While it appears to have occurred both repeatedly and independently in different lineages, the generative process is enigmatic (*Brunet and King, 2017*; *Richter et al., 2018*; *Shapiro, 1998*); moreover, we cannot directly observe the transmission process of extant organisms, including HS-3. Nonetheless, in this study, we found unique characteristics of HS-3 that have some implications regarding the emergence of multicellularity.

The remarkable aspect of HS-3's life-cycle is the regulated dimorphism between the first growth stage (filamentous cells showing liquid-crystal-like self-organization) and the second stage (coccobacillus cells). The first stage suggests that liquid-crystal-like self-organization is involved in the emergence of multicellularity, which has not been reported before. The liquid-crystal behavior of microscopic cell populations is recognized as the spontaneous formation of intrinsic two-dimensional patterns when motile, rod-shaped cells proliferate in thin layers at a certain density. Typical two-dimensional nematic patterns are characterized by asymmetric vortex structures and anisotropic domains adjacent to specific types of topological defects (*Duclos et al., 2017*; *Majumdar et al., 2014*; *Sanchez et al., 2012*); this was also observed for HS-3 colony on solid surface (*Figure 3—figure supplement 1*). The rise of the fluid dynamics subdiscipline in physics led to a number of theoretical and experimental approaches for cell behaviors in the last two decades, where researchers observed liquid-crystal orderings in cell suspension of neuronal cells, fibroblasts, melanocytes, osteoblasts, lipocytes, and several laboratory model bacteria such as *Escherichia coli* and *Bacillus subtilis* (*Doostmohammadi et al., 2018*). In ordinary bacteria as observed in *E. coli* growing on agar, the nematic features are only sustained for short periods (*Dell'Arciprete et al., 2018*; *Yaman et al., 2019*). In contrast, HS-3 is unique in its ability to maintain the two-dimensional sheet structure for a prolonged period, which we consider to be one of the prerequisites for HS-3 to establish its multicellular behavior.

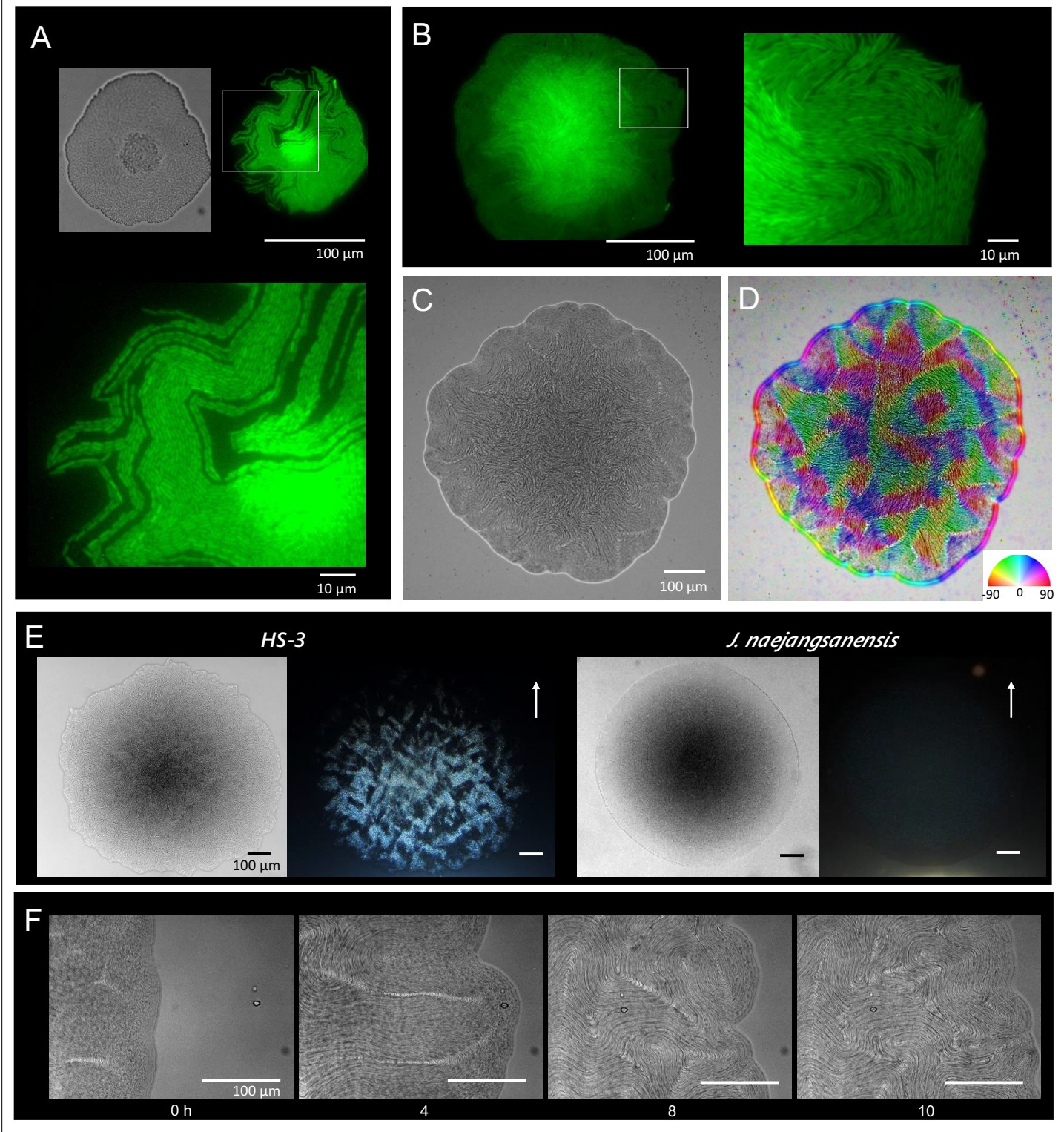

**Figure 3.** Colony morphogenesis and liquid-crystal-like ordering. (**A**) A microcolony showing a single layer of cells. Differential interference contrast (DIC) (top left) and green fluorescent protein (GFP) (top right) images and an enlarged image (bottom) of the area indicated by the white box. (**B**) A young colony (larger than A) with a single layer of cells at the edge. Enlarged image of the area indicated by the white box (right). (**C**) Anisotropic pattern of the bottom cell layer of a single colony. (**D**) The color mapping to show the orientation of cells. The anisotropic pattern is a typical characteristics in a nematic phase (*Doostmohammadi et al., 2018*). (**E**) Anisotropic optical texture of colonies of HS-3 (left) and *Jeongeupia naejangsanensis* (right), the close relative strain. DIC images (left) and diagonally illuminated images (right). Arrows indicate the direction of light

*Figure 3 continued on next page*

*Figure 3 continued*

illumination. (**F**) Time-lapse images of liquid-crystal-like pattern formation on the edge of a growing colony. Time (hr) after chase is indicated. The original video is available as *Video 1*.

The online version of this article includes the following figure supplement(s) for figure 3:

**Figure supplement 1.** Emergence of analogous nematic patterns on the colony edge.

This prolonged nematic status enables HS-3 to make its colony growth a two-phase life-cycle. Two distinct life-cycles are seen in extant single-celled bacteria such as spore formation in *B. subtilis* and asymmetric cell division of *Caulobacter crescentus* (*Casadesús and Low, 2013*; *Dubnau and Losick, 2006*). More generally, bacteria produce differentiated subpopulations using a variety of genetic (e.g., phase variation) or nongenetic (e.g., bistability) mechanisms (*Casadesús and Low, 2013*; *Dubnau and Losick, 2006*), by which they can achieve division of labor or hedge their bets in the event of unpredictable future conditions (*Malik and Rosenberg, 2009*). Although these dichotomic behaviors at population level underlie in multicellular life, bacteria still require some more process to acquire and establish distinctive multicellular structures. In the case of HS-3, we observed that internal growth of coccobacilli was attenuated until the colony of fibrous cells matured, and this regulation is likely a factor that enabled the establishment of multicellular-like structure.

Another important condition to support HS-3's multicellular-like behavior appears to be the dynamic water environment in the cave, which we believe is the key to explain a potential evolutionary path toward HS-3's multicellularity. The recurrent dynamism of water flow could have allowed each colony on the wall (i.e., a collective of genetically homogenous population) to undergo Darwinian natural selection. This assumption gives an attractive explanation as to how a genetically homogeneous single-cellular organism might have experienced natural selection through a benefit during the early stage of the population characteristics, and supports the 'Ecological Scaffolding' theory (*Black et al., 2020*).

In summary, the present study underpins the aforementioned theories and concepts about the transition from unicellular to multicellular life. HS-3 is an example of an extant prokaryote with a unique multicellular behavior that is based on liquid-crystal characteristics and timely regulated development, and potentially involved recurrent dynamism in its cave environment in the emergence of multicellularity.

## Materials and methods
### Strains and culture media

Strain HS-3 was isolated in 2008 from percolating water on the surface of a limestone cave wall in 'Seryukutsu' (Blue Dragon) cave in the Hirao karst plateau (33°46′00.5″ N 130°55′02.7″ E) in northern Fukuoka Prefecture, Japan (*Figure 2—figure supplement 1*). The sampling site on the limestone wall was situated close to the surface of an underground river. Water dripping from the wall was collected using a sterile plastic tube and the pH and temperature were measured on site. Samples were then stored at 4°C for several days until analysis. The water sample was diluted with phosphate-buffered saline (PBS, pH 7.0) to inoculate R2A (0.5 g yeast extract, 0.5 peptone, 0.5 casamino acids, 0.5 glucose, 0.3 $K_2HPO_4$, 0.05 $MgSO_4$ per liter of deionized water) agar plates containing Nile red (Sigma-Aldrich,



**Video 1.** The edge of a colony.
https://elifesciences.org/articles/71920/figures#video1

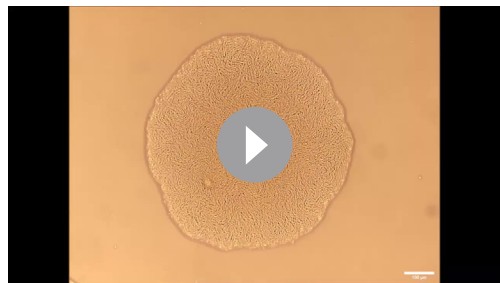

**Video 2.** Emergence of coccobacillus cells.
https://elifesciences.org/articles/71920/figures#video2

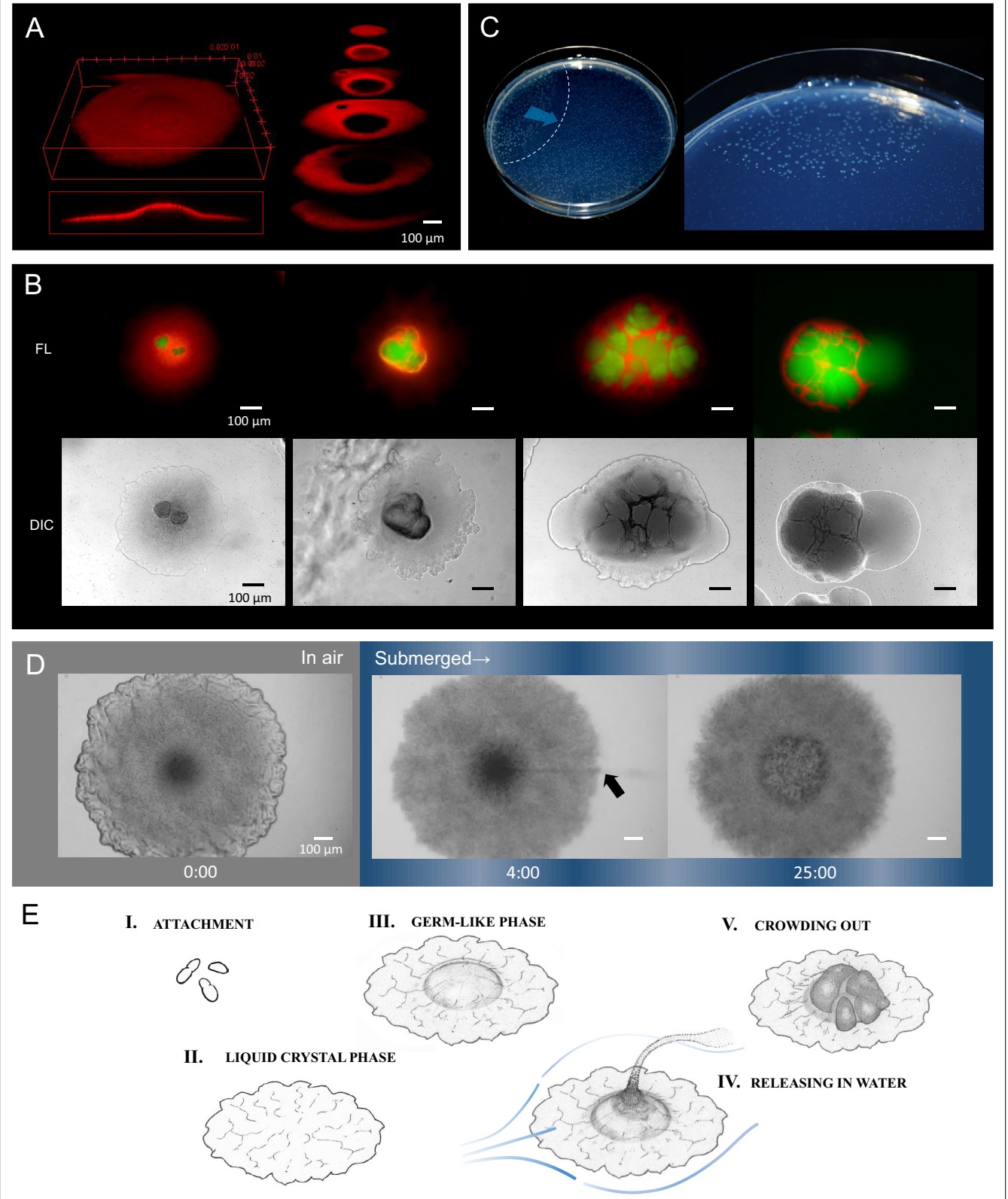

**Figure 4.** Internally growing cells and crowding out. (**A**) Three-dimensional image of a 4-day-old colony reconstructed from 100 Z-axis slices with a total height of 168 µm. The whole Nile-red image (left) and its cross-sections (right). (**B**) Five-day-old, mature colonies showing internal growth at the center of the colony. Two right panels show the 'crowding out'. (**C**) Colony-crowding influences adjacent colonies by inducing a 'chain reaction'. The direction of the chain reaction and its front line are indicated by the blue arrow and the dotted line, respectively. More details of a crowding colony

*Figure 4 continued on next page*

*Figure 4 continued*

are shown in *Video 2*. (**D**) Time-lapse images of a mature colony in air (left panel) and after being submerged in water (middle and right panels). The arrow indicates the waterborne coccobacillus cells released from the colony. The colonies were grown for 5 days on agar before the time-lapse analysis. The time (min) after chase is shown. Bars = 100 µm. The original video is available as *Video 3*. (**E**) Illustration of the potential multicellular life-cycle of HS-3. (I) Coccobacillus cells attached to a new solid surface. (II) The cells proliferate to some extent, generating a liquid-crystal texture. (III) The center area becomes raised due to clustering of internal cells, leading to the germ-like phase. (IV) A waterflow releases coccobacilli. (V) In the absence of a waterflow, outward growth of the internal coccobacillus cells results in 'crowding out'.

The online version of this article includes the following figure supplement(s) for figure 4:

**Figure supplement 1.** Reversibility between filament and coccobacillus cells.

Missouri, USA) at 0.5 mg per liter. The HS-3 strain was originally isolated as a Nile red-positive strain, indicating that it was a lipid producer. The isolated HS-3 strain was precultured in R2A liquid medium in a test tube overnight at 24°C in a reciprocal shaker at 100 rpm. Then, the culture was diluted with PBS and inoculated on a plate containing R2A and 1.5% agar for taxonomic and microscopic analysis. The close relatives, *Jeongeupia naejangsanensis* KCTC 22633[T] and *J. chitinilytica* KCTC 23701[T] were purchased from the Korean Collection for Type Cultures, Taejon, Korea. Two closely related bacteria, *Andreprevotia chitinilytica* NBRC106431[T] and *Silvimonas terrae* NBRC100961[T], were purchased from the National Bio-Resource Center, Kisarazu, Japan. Those strains were cultured in R2A medium at 24 or 30°C on agar, or in a reciprocal shaker at 100 rpm for liquid cultures.

## Taxonomic identification of strain HS-3

The Gram reaction was performed using a Gram Staining Kit (Sigma). Culture media and conditions for taxonomic comparisons were essentially the same as those used for the reference strains, *J. naejangsanensis* KCTC 22633[T] (*Yoon et al., 2010*) and *J. chitinilytica* KCTC 23701[T] (*Chen et al., 2013*). Growth at various temperatures (4, 10, 16, 18, 20, 22, 24, 26, 28, 30, 32, 34, and 40°C) and pH (3.0–11.0 at intervals of 0.5) was performed in nutrient broth (Becton Dickinson, New Jersey, USA). The pH was adjusted by using sodium acetate/acetate, phosphate, and $Na_2CO_3$ buffers. Growth at different NaCl concentrations (0, 0.5, 1.0, 2.0, 3.0, 4.0, and 5.0%, wt/vol) was also performed in tryptic soy broth prepared according to the formula used for the Difco medium, except that NaCl was excluded. Growth under anaerobic conditions was performed using an anaerobic chamber with a gas exchange kit (Anaeropack; Mitsubishi Gas Chemical, Tokyo, Japan). Catalase and hydrolysis of casein, gelatin, starch, and Tween 20, 40, 60, and 80 were determined as described by *Cowan and Steel, 1965*. Additionally, assimilation of Tween 20, 40, 60, and 80 as sole carbon sources was tested in M9 medium (12.8 g $Na_2HPO_4 \cdot 7H_2O$, 3 g $KH_2PO_4$, 0.5 g NaCl, and 1 g $NH_4Cl$ per liter of deionized water containing 2 mM $MgSO_4$ and 0.1 mM $CaCl_2$). Susceptibility to antibiotics was tested by placing antibiotic-impregnated discs on agar plates seeded with the test strain. The antibiotics tested were kanamycin, gentamycin, erythromycin, fosfomycin, trimethoprim, novobiocin, streptomycin, chloramphenicol, ampicillin, and tetracycline. Utilization of various substrates and biochemical properties were tested by using API 20NE, API 50CH, API ZYM kits (bioMerieux, Marcy l'Etoile, France). Major types of quinones were determined at Techno Suruga, Shizuoka, Japan, using reverse-phase high-performance liquid chromatography (HPLC) (*Komagata and Suzuki, 1988*). Cellular fatty acid composition was also determined at Techno Suruga using the MIDI/Hewlett Packard Microbial Identification System (*Sasser, 1990*). The strain was deposited at the National Bioresource Center, Kisarazu, Japan as *Jeongeupia* sp. HS-3 NBRC 108274[T] (=KCTC 23748[T]).

Genomic DNA was extracted using a Qiagen kit (Qiagen, Hilden, Germany), and the 16S rRNA gene was amplified using the primer set of 27 f and 1525 r (*Lane, 1996*). The PCR product was sequenced (Applied Biosystems 3730xl DNA Analyzer, Applied Biosystems, USA), and the DNA G + C content was measured by Techno Suruga Inc using HPLC (*Katayama-fujimura et al., 2014*). A phylogenetic tree was constructed using the neighbor-joining method (*Saitou and Nei, 1987*)

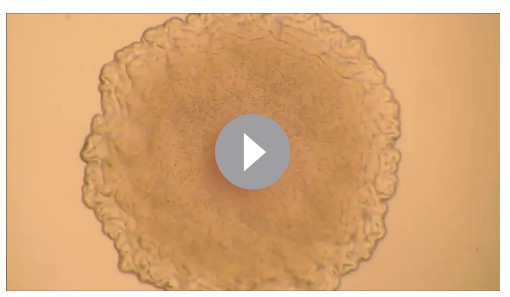

**Video 3.** Waterborne coccobacillus cells from a colony.
https://elifesciences.org/articles/71920/figures#video3

implemented in the Clustal W software package (*Thompson et al., 1994*) using sequences of closely related bacteria in GenBank that were selected by BLAST (*Altschul et al., 1997*).

Genomic DNA for genome sequencing was extracted using a NucleoSpin kit (Macherey-Nagel, Düren, Germany). Complete sequencing was performed at the Oral Microbiome Center, Takamatsu, Japan, using a combination of long-read sequencing performed using a Nanopore system (Oxford Nanopore Technologies, Oxford, UK) and short-read sequencing performed using a DNBSEQ system (MGI Tech, Shenzhen, China). The sequence data were analyzed using Lasergene software (DNASTAR) and annotations were performed by DFAST (https://dfast.nig.ac.jp/). The complete genome sequence and a plasmid sequence have been registered in the DDBJ database (https://www.ddbj.nig.ac.jp/) as AP024094 and AP024095, respectively.

## GFP recombination

Competent cells of strain HS-3 were prepared using general laboratory protocols (*Sambrook and Russel, 2001*). Briefly, cells cultured in liquid R2A medium for 48 hr at 24°C were inoculated into a fresh R2A medium at a dilution of 20 times. Then, the cells were cultured until the $OD_{600\,nm}$ reached 0.5. The cells were harvested by centrifugation and washed two to three times with sterile water and 10% glycerol, with a gradual increase in the concentration of the suspended cells. Finally, the cells were suspended in 10% glycerol, distributed to tubes, and stored at −80°C. The broad-host-range plasmid, pBBR1MCS2-pAmp-EGFP (*Kovach et al., 1994*) (Nova Lifetech, Hong Kong, China), was transformed into competent cells of HS-3 by electroporation under the following conditions. A mixture containing 150 ng of the plasmid and 100 µl of the cell suspension was placed in a cuvette with a 2.0 mm gap and a voltage of 1.8 kV was applied. The suspension was then mixed with 1 ml of R2A medium and incubated for 2 hr at 24°C with shaking. The transformants were selected on an R2A agar plate containing kanamycin (25 µg/ml).

## Light and fluorescence microscopy

Cells were cultured in R2A medium either in a test tube containing liquid medium, or on a 1.5% agar or 3.0% gellan gum plate. The liquid-cultured cells were then observed on a glass slide under a microscope (Nikon Eclipse 80i, Nikon Inc, Japan). For time-lapse video microscopy, colonies grown on agar were observed directly on a Petri dish under the microscope using a ×10 or ×40 objective. Nile-red fluorescent images were obtained for the colonies that were inoculated and grown as described above. The colonies were observed under the microscope with filters for excitation (525–540 nm) and emission (>565 nm) wavelengths. To obtain three-dimensional images of a colony, a small piece of agar was cut out from an agar plate and placed on a glass slide. The slide was then observed under a confocal laser scanning microscope (Leica Microsystems TCS SP8, Leica Microsystems, Wetzlar, Germany) with filters for excitation (488–552 nm) and emission (560–620 nm) wavelengths and a ×10 objective. The images were processed using ImageJ 1.52 software (https://imagej.nih.gov/ij/).

## Electron microscopy

A gel slice of a 48-hr-old colony was cut out from a Petri dish and soaked overnight in a 4% paraformaldehyde solution in 100 mM phosphate buffer (pH 7.4), and then soaked for 1 hr twice in 20 mM phosphate buffer (pH 7.0). The slice was then dried in a desiccator at room temperature overnight, and then dried under reduced pressure (−20 kPa) overnight. The slice was mounted on an appropriate grid and examined under a scanning electron microscope (JSM-6510LA, JEOL, Tokyo, Japan). Transmission electron microscopy (TEM) was performed at Hanaichi Ultrastructure Research Institute, Nagoya, Japan. The liquid-cultured cells were mounted on a grid and washed with distilled water at room temperature. Excess liquid was removed and the specimen was negatively stained using 2% uranyl acetate before being observed by TEM (JEM1200EX, JEOL).

## Statistical analysis

All experiments were conducted in triplicate. Values represent the mean ± standard error of the mean of three independent experiments.

## Acknowledgements

We thank the following people for their support with this study: Dr. Masahiro Nakano for discussion on aspects related to physics. Ms. Aya Ohta for isolating the bacterium, Ms. Satoko Nakanomori and Mr. Sakae Fukase for phenotype characterization, Mr. Hiroyuki Tanaka for SEM analysis, and Dr. Vishal Gor for his critical manuscript reading and English editing.

## Additional information

### Funding

No external funding was received for this work.

### Author contributions

Kouhei Mizuno, Conceptualization, Data curation, Formal analysis, Supervision, Investigation, Visualization, Methodology, Writing – original draft, Project administration, Writing – review and editing; Mais Maree, Data curation, Formal analysis, Methodology, Writing – original draft, Writing – review and editing; Toshihiko Nagamura, Investigation, Methodology, Writing – original draft, Writing – review and editing; Akihiro Koga, Investigation, Methodology; Satoru Hirayama, Kenji Tanaka, Investigation, Methodology, Writing – original draft; Soichi Furukawa, Conceptualization, Investigation, Methodology; Kazuya Morikawa, Conceptualization, Supervision, Investigation, Methodology, Writing – original draft, Writing – review and editing

### Author ORCIDs

Kouhei Mizuno (ID) http://orcid.org/0000-0002-2070-8669
Satoru Hirayama (ID) http://orcid.org/0000-0003-0965-8733
Kazuya Morikawa (ID) http://orcid.org/0000-0002-1503-7151

### Decision letter and Author response

Decision letter https://doi.org/10.7554/eLife.71920.sa1
Author response https://doi.org/10.7554/eLife.71920.sa2

## Additional files

### Supplementary files

• Supplementary file 1. Characteristics that distinguish strain HS-3 from its close phylogenetic relatives.

• Transparent reporting form

### Data availability

The complete genome sequence and a plasmid sequence have been registered in the DDBJ database (https://www.ddbj.nig.ac.jp/) as AP024094 and AP024095, respectively.

The following previously published datasets were used:

| Author(s) | Year | Dataset title | Dataset URL | Database and Identifier |
|-----------|------|---------------|-------------|-------------------------|
| Mizuno K, Maree M, Morikawa K | 2020 | Jeongeupia sp. HS-3 plasmid pJHS3 DNA, complete sequence | https://www.ncbi.nlm.nih.gov/nuccore/AP024095.1 | NCBI Nucleotide, AP024095.1 |
| Mizuno K, Maree M, Morikawa K | 2020 | Jeongeupia sp. HS-3 DNA, complete genome | https://www.ncbi.nlm.nih.gov/nuccore/AP024094.1 | NCBI Nucleotide, AP024094.1 |

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
