## [Editor Report]

This article reports the discovery of an unusual form of bacterial multicellularity – an organism that can exist in dense, filamentous multicellular structures and clusters of coccobacillus daughter cells. Experiments that mimic the periodic immersion that the bacteria experience in their natural cave environment suggest that water immersion plays a role in these life-cycle dynamics. This work, while rather qualitative, will likely attract great interest from a diverse range of scientists working on multicellularity, the biophysics of cell packing, and geobiological problems.

---

## [Decision Letter]

**Decision letter after peer review:**

Thank you for submitting your article "Novel multicellular prokaryote discovered next to an underground stream" for consideration by *eLife*. Your article has been reviewed by 2 peer reviewers, one of whom is a member of our Board of Reviewing Editors, and the evaluation has been overseen by Gisela Storz as the Senior Editor. The reviewers have opted to remain anonymous.

Essential revisions:

1) The reviewers are enthusiastic about the discovery but think the description of its significance is lacking.

They suggest reframing it, not as an example of a missing 'group phase' (which isn't in fact missing, and this paper is not a very compelling argument in any case since it is not on a lineage that has ultimately evolved complex multicellularity), but rather as a new form of bacterial multicellularity.

'Ecological scaffolding' is a hot topic in the field- the idea that ecology gives rise to the first multicellular life cycles, which then are co-opted by developmental processes (see Black, Bourrat and Rainey, 2020). We suggest to focus on (a) the novel biology of a new type of bacterial multicellularity, which is significant since we know that bacteria were forming multicellular colonies back 3.42 billion years ago (Calvazzi 2021), with the first honest-to-goodness multicellular lineages (i.e., cyanobacteria), and (b) how reliable ecological cycles (ecological scaffolding) may have driven the evolution of a new type of bacterial multicellularity in these caves.

(2) The authors should place their work more clearly in the context of recent work on the advantages of multicellular forms in other evolutionary contexts, where response to environmental pressures are thought to drive adoption of certain strategies.

[Editors’ note: further revisions were suggested prior to acceptance, as described below.]

Thank you for resubmitting your work entitled "Novel multicellular prokaryote discovered next to an underground stream" for further consideration by *eLife*. Your revised article has been evaluated by Gisela Storz (Senior Editor) and a Reviewing Editor. We apologize for the long delay in furnishing this decision.

The manuscript has been improved but there are some remaining issues that need to be addressed, as outlined below:

Summary of reviewers' comments:

The reviewers appreciate the changes you have made in response to the first round of review but are of the opinion that the way in which your discovery is framed is not particularly well justified and may even detract from the main message of your paper.

Our main concern is that the paper is framed about gaps in our knowledge about how multicellularity evolves, but the paper does not actually do a good job addressing these gaps. We don't actually have any information about what its unicellular ancestors looked like or behaved, how selection acted on early multicellular stages, what transitional steps in its evolution were like, etc. The authors cannot address these questions, because they just have a single contemporary species. And these questions have been addressed in a large literature that the authors do not cite- classically involving the volvocine green algae, and experimental evolution work in yeast, *Chlamydomonas*, and *Pseudomonas*.

What we do have in this paper is a very nice example of bacterial multicellularity that is different from other examples of bacterial multicellularity. And, importantly, it appears that multicellularity, in this case, is adaptive because of the periodic flooding behavior of the cave, which allows for groups to form, generate, and then periodically disperse. We thought that the environment in which these microbes presumably evolved in might provide some neat insight into their evolution, but even this claim must be made lightly since we do not know how these organisms actually evolved. Specifically, we suggested that periodic flooding might act as an ecological scaffold, creating an early life cycle in which groups form and disperse based purely on the abiotic environment. This connects nicely to recent theoretical work on scaffolding, but it seems from the paper that the authors' do not understand what scaffolding is or how their system can be interpreted in this context.

We think that the introduction needs to be reworked, as in its present form it appears as a meandering list of some of what is known about the early evolution of multicellularity, but it does not review the topic well, build to a critical point or identify gaps in the knowledge that the paper does a good job of addressing. This revised version is, in our opinion, not as good as the first one, as more information has been added but it just makes the arguments about the evolution of multicellularity less coherent.

In the same vein, we are concerned about Figure 1: first, ecological scaffolding is not widely accepted as a necessary step in the transition to multicellularity. In fact, we do not even have a single good example of it happening. That's actually why we suggested it for this paper in the last review, and we think that this paper might be the best example yet of an idea that so far has just been shown to work in models. Though, of course, we don't know if scaffolding played a role in the evolution of HS3, it seems plausible given the environment. Recurrent dynamism (which is their term for what ecological scaffolding is- these are the same exact argument!), dichotomic cell response, and trade-offs are not all hypothetical conditions for the origin of multicellularity. Scaffolding is, but dichotomic cell response and trade-offs are not conceptual foundations for how multicellularity evolves. A dichotomic cell response will usually be a trait that can be adaptive in a multicellular life cycle, usually because it provides a benefit through division of labor/differentiation. This is really just saying 'cellular differentiation' in using the authors' own unique terminology. And trade-offs are a part of life that constrains how evolution works- a universal law of nature that applies to all of evolutionary biology. Critically, these are three different kinds of ideas, one a conceptual scheme for how multicellularity evolves, one a description of how cells behave, and one a basic feature of all biological evolution.

We do not understand why they are in the figure as 'hypothetical conditions for the emergence of multicellularity'.

We believe that this paper stands on its own two feet purely as an example of fascinating organismal biology, a new kind of multicellular bacteria found in caves! That is important in its own right.

We thus urge the authors to go with their strengths. Talk about the discovery of a fascinating new multicellular organism. Perhaps include a bit about what this teaches us about evolution in the discussion. But to use this as a foundation for the paper appears to us as a mistake, as they fail to clearly identify the gaps in the knowledge of how their work addresses them.

---

## [Author Response]

Essential revisions:1) The reviewers are enthusiastic about the discovery but think the description of its significance is lacking.They suggest reframing it, not as an example of a missing 'group phase' (which isn't in fact missing, and this paper is not a very compelling argument in any case since it is not on a lineage that has ultimately evolved complex multicellularity), but rather as a new form of bacterial multicellularity.'Ecological scaffolding' is a hot topic in the field- the idea that ecology gives rise to the first multicellular life cycles, which then are co-opted by developmental processes (see Black, Bourrat and Rainey, 2020). We suggest to focus on (a) the novel biology of a new type of bacterial multicellularity, which is significant since we know that bacteria were forming multicellular colonies back 3.42 billion years ago (Calvazzi 2021), with the first honest-to-goodness multicellular lineages (i.e., cyanobacteria), and (b) how reliable ecological cycles (ecological scaffolding) may have driven the evolution of a new type of bacterial multicellularity in these caves.(2) The authors should place their work more clearly in the context of recent work on the advantages of multicellular forms in other evolutionary contexts, where response to environmental pressures are thought to drive adoption of certain strategies.

We have revised the Introduction, Discussion, and Figure 1 to appropriately contextualize this study as a new form of bacterial multicellularity (described in detail below).

[Editors’ note: further revisions were suggested prior to acceptance, as described below.]

Summary of reviewers' comments:The reviewers appreciate the changes you have made in response to the first round of review but are of the opinion that the way in which your discovery is framed is not particularly well justified and may even detract from the main message of your paper.Our main concern is that the paper is framed about gaps in our knowledge about how multicellularity evolves, but the paper does not actually do a good job addressing these gaps. We don't actually have any information about what its unicellular ancestors looked like or behaved, how selection acted on early multicellular stages, what transitional steps in its evolution were like, etc. The authors cannot address these questions, because they just have a single contemporary species. And these questions have been addressed in a large literature that the authors do not cite- classically involving the volvocine green algae, and experimental evolution work in yeast, Chlamydomonas, and Pseudomonas.

We introduced and cited previous works in Introduction and Discussion and amended the Figure 1.

What we do have in this paper is a very nice example of bacterial multicellularity that is different from other examples of bacterial multicellularity. And, importantly, it appears that multicellularity, in this case, is adaptive because of the periodic flooding behavior of the cave, which allows for groups to form, generate, and then periodically disperse. We thought that the environment in which these microbes presumably evolved in might provide some neat insight into their evolution, but even this claim must be made lightly since we do not know how these organisms actually evolved. Specifically, we suggested that periodic flooding might act as an ecological scaffold, creating an early life cycle in which groups form and disperse based purely on the abiotic environment. This connects nicely to recent theoretical work on scaffolding, but it seems from the paper that the authors' do not understand what scaffolding is or how their system can be interpreted in this context.

We carefully considered the concept of ecological scaffolding in the context of this study, and reorganized the whole context of this manuscript to focus on the bacterial multicellularity evolved in the cave removing extra information as suggested by the reviewers.

We think that the introduction needs to be reworked, as in its present form it appears as a meandering list of some of what is known about the early evolution of multicellularity, but it does not review the topic well, build to a critical point or identify gaps in the knowledge that the paper does a good job of addressing. This revised version is, in our opinion, not as good as the first one, as more information has been added but it just makes the arguments about the evolution of multicellularity less coherent.In the same vein, we are concerned about Figure 1: first, ecological scaffolding is not widely accepted as a necessary step in the transition to multicellularity. In fact, we do not even have a single good example of it happening. That's actually why we suggested it for this paper in the last review, and we think that this paper might be the best example yet of an idea that so far has just been shown to work in models. Though, of course, we don't know if scaffolding played a role in the evolution of HS3, it seems plausible given the environment. Recurrent dynamism (which is their term for what ecological scaffolding is- these are the same exact argument!), dichotomic cell response, and trade-offs are not all hypothetical conditions for the origin of multicellularity. Scaffolding is, but dichotomic cell response and trade-offs are not conceptual foundations for how multicellularity evolves. A dichotomic cell response will usually be a trait that can be adaptive in a multicellular life cycle, usually because it provides a benefit through division of labor/differentiation. This is really just saying 'cellular differentiation' in using the authors' own unique terminology. And trade-offs are a part of life that constrains how evolution works- a universal law of nature that applies to all of evolutionary biology. Critically, these are three different kinds of ideas, one a conceptual scheme for how multicellularity evolves, one a description of how cells behave, and one a basic feature of all biological evolution.We do not understand why they are in the figure as 'hypothetical conditions for the emergence of multicellularity'.

We amended Figure 1 and removed some extra concepts in Introduction and Discussion.